# Developing a Prediction Model of Demolition-Waste Generation-Rate via Principal Component Analysis

**DOI:** 10.3390/ijerph20043159

**Published:** 2023-02-10

**Authors:** Gi-Wook Cha, Se-Hyu Choi, Won-Hwa Hong, Choon-Wook Park

**Affiliations:** 1School of Science and Technology Acceleration Engineering, Kyungpook National University, Daegu 41566, Republic of Korea; 2School of Architectural, Civil, Environmental and Energy Engineering, Kyungpook National University, Daegu 41566, Republic of Korea; 3Industry Academic Cooperation Foundation, Kyungpook National University, Daegu 41566, Republic of Korea

**Keywords:** demolition-waste-generation rate, hybrid model, machine learning, principal component analysis, waste management

## Abstract

Construction and demolition waste accounts for a sizable proportion of global waste and is harmful to the environment. Its management is therefore a key challenge in the construction industry. Many researchers have utilized waste generation data for waste management, and more accurate and efficient waste management plans have recently been prepared using artificial intelligence models. Here, we developed a hybrid model to forecast the demolition-waste-generation rate in redevelopment areas in South Korea by combining principal component analysis (PCA) with decision tree, k-nearest neighbors, and linear regression algorithms. Without PCA, the decision tree model exhibited the highest predictive performance (*R*^2^ = 0.872) and the k-nearest neighbors (Chebyshev distance) model exhibited the lowest (*R*^2^ = 0.627). The hybrid PCA–k-nearest neighbors (Euclidean uniform) model exhibited significantly better predictive performance (*R*^2^ = 0.897) than the non-hybrid k-nearest neighbors (Euclidean uniform) model (*R*^2^ = 0.664) and the decision tree model. The mean of the observed values, k-nearest neighbors (Euclidean uniform) and PCA–k-nearest neighbors (Euclidean uniform) models were 987.06 (kg·m^−2^), 993.54 (kg·m^−2^) and 991.80 (kg·m^−2^), respectively. Based on these findings, we propose the k-nearest neighbors (Euclidean uniform) model using PCA as a machine-learning model for demolition-waste-generation rate predictions.

## 1. Introduction

Owing to recent developments in artificial intelligence, machine-learning models have been widely studied for waste generation prediction. Machine learning has been successfully applied in developing waste generation prediction models, owing to its excellent ability to model complex mechanisms [1,2]. Information on waste generation is essential in waste management strategies, such as the planning of landfill spaces, calculation of levies on pollution sources or polluters and of subsidies for recycling, and establishment of corporate waste management policies [3].

With the rapid growth of urban populations, waste management has become an important issue for urban quality of life [4,5]. Waste treatment imposes social costs and environmental burdens, affecting carbon emissions, traffic congestion, and air quality. Proper waste management is therefore essential in constructing sustainable and comfortable urban environments [6,7,8]. Construction and demolition waste (CDW) is a key by-product of urbanization and substantially contributes to environmental degradation [9,10,11]. The construction industry is considered harmful to the environment because it is a major consumer of natural resources, materials, and energy; furthermore, the demolition of buildings at the end of their service life generates substantial amounts of waste [12,13]. CDW accounts for 35 to 40% of waste generation worldwide [14], 36% in the EU, and 67% in the United States [15]. Additionally, 70 to 90% of CDW is attributed to demolition waste [16,17]. As a result of its environmental effects and increasing generation, CDW presents a major challenge for the construction industry. Due to its social and environmental impacts, in addition to a huge amount of waste emissions, CDW has been the subject of much research, particularly during periods of rapid development [18,19].

Attempts have been made to develop predictive models for waste management in relation to waste generation, and several machine-learning algorithms have been utilized. For the period 2004 to 2019, Abdallah et al. [20] reported that the most commonly used machine-learning algorithms for the development of artificial intelligence models related to waste management were artificial neural networks (ANNs), support vector machines (SVMs), linear regression (LR) analysis, decision trees (DTs), and genetic algorithms (GAs), while random forest (RF) and k-nearest neighbor (KNN) algorithms have also been used. The artificial intelligence models used in different studies, applying these algorithms individually, exhibited variable predictive performances even when developed using the same algorithm. This variability is due to differences between the studies in the data used (for instance, in sample size and input variable characteristics), the particular advantages and disadvantages of each machine-learning algorithm, and the selection of hyperparameters, factors that affect and limit machine-learning models’ predictive performance. The advantages and disadvantages of each machine-learning algorithm constrain the potential improvements in performance that they can achieve.

To overcome such limitations, research has been conducted on the development of hybrid models combining various artificial intelligence systems; such hybrid models show better predictive performance than single algorithms. For example, a hybrid model to predict solid waste generation, applying the wavelet denoising method and partial least squares to SVM, achieved better predictive performance than the SVM model [21,22]. When developing a CDW-prediction model, applying a gray model improved the predictive performance of a support vector regression (SVR) model [14]. A hybrid model combining a GA and ANN to predict municipal solid waste achieved significantly better predictive performance than the ANN alone, raising the *R*^2^ from 0.13 to 0.78 [23]. A categorical principal component analysis (CATPCA) hybrid model using six input variables and generating six principal components (PCs) was applied to ANN, SVR, and RF to predict the demolition-waste-generation rate (DWGR). The hybrid PCA–SVR model achieved significantly better performance than SVR alone (SVR, *R*^2^ = 0.007; hybrid, *R*^2^ = 0.594) [24]. 

Based on these findings, the use of hybrid models has enabled continuous improvement in waste-generation-predictive performance and provides a means to develop effective artificial intelligence models for waste generation prediction. To further improve this, we aimed to develop a hybrid machine-learning model to predict DWGR in redevelopment areas in South Korea, by testing machine-learning algorithms appropriate for the data, selecting optimal hyperparameters, and combining them with artificial intelligence approaches. We focused on the DWGR because as a tool for developing waste management strategies, the waste generation rate has been widely applied in CDW management and research [17,25,26,27]. 

The main purpose of this study is to develop optimal ML models to forecast DWGR. More specifically, the detailed purpose is not only to develop a DWGR prediction model, but also to suggest ways to improve the prediction performance. In order to achieve this research purpose, this study presented a new approach to PCA utilization and developed a hybrid model with excellent predictive performance.

## 2. Methods and Materials

Figure 1 summarizes the research workflow. We first constructed a dataset by collecting DWGR data for 160 buildings in redevelopment areas in South Korea. LR, KNN, and DT algorithms were considered for developing the models, and the optimal hyperparameters were selected for each machine-learning algorithm. To improve model performance, hybrid models were developed by applying PCA. Model validation was conducted using leave-one-out cross-validation, and performance was evaluated using mean squared error (*MSE*), root mean square error (*RMSE*), coefficient of determination (*R*^2^), and mean absolute error (*MAE*).

### 2.1. Data Collection and Preprocessing

We collected information on building characteristics (location, structure, usage, wall type, roof type, gross floor area (GFA, in m^2^), and number of floors) by directly surveying each building before demolition in two South Korean cities’ redevelopment areas, namely Project A in Daegu (35.88 N latitude, 128.61 E longitude) and Project B in Busan (35.87 N latitude, 128.63 E longitude). The study areas were urban regeneration districts, consisting largely of aged buildings that are primarily low-rise. As Korea plans to demolish old buildings via urban regeneration projects, the amount of demolition waste is expected to increase significantly. Hence, the current study aims to provide useful information to aid governments and other stakeholders in their waste management approach. 

Following demolition, DWGR information was obtained from truck transport information. Mean DWGR differed clearly with location, usage, structure, wall type, and roof type (Table 1), differing significantly between Projects A (741.6 kg·m^−2^) and B (1238.9 kg·m^−2^). The mean DWGR of mixed-use (residential and commercial) buildings was approximately 30% higher than that of residential buildings. These factors were therefore used as the input variables to predict DWGR. For most buildings, the GFA and DWGR were within 300 m^2^ and 1800 kg·m^−2^, respectively, because they were mostly old low-rise buildings in redevelopment areas. DWGR was defined as follows:(1)$$DWGR_{i}=\frac{\sum A\;of\;building_{i}}{GFA\;of\;building_{i}}$$
where $DWGR_{i}$ is the demolition-waste-generation rate (kg·m^−2^) and *A* is the waste generated (kg) for building i.

To improve the machine-learning model performance by using a uniform data scale, the data were first normalized as follows:(2)$$x_{norm}=\frac{x-x_{min}}{x_{max}-x_{min}}$$
where x is the data value, and $x_{max}$ and $x_{min}$ are the maximum and minimum values, respectively. 

### 2.2. Applied Machine-Learning Algorithms

Our data were low dimensional, comprising few input variables (five categorical and two numerical). We first applied the DT algorithm, which can be applied regardless of the input variable type. Although the KNN algorithm is suitable for low-dimensional data [28], it achieved poor predictive performance (*R*^2^ = 0.51) in a prior study [29], and it has rarely been applied in studies of waste generation. For these reasons, we selected it for use here. 

The LR algorithm has been widely used in studies of waste generation, with variable predictive performance [21]. As it assumes linear relationships, however, this method is not optimal for modelling highly non-linear data [21,28]. Nonetheless, we assumed that it would be useful for testing whether its performance can be improved by applying PCA.

#### 2.2.1. Principal Component Analysis

PCA is a multivariate statistical method that reduces the complexity introduced by the inclusion of multiple variables [30] by reducing many variables into fewer variables (PCs) that explain the variance in the data. PCA converts the input variables into independent and linear compound PCs [31], which can then be used as input variables for artificial intelligence model development. Noori et al. (2008) [31] attempted to develop a hybrid PCA–SVM model for predicting solid waste generation, comparing an SVM model utilizing thirteen input variables with a hybrid model using six PCs: the hybrid model did not effectively improve performance (SVM alone, *R*^2^ = 0.78; hybrid: *R*^2^
*=* 0.75). Applying PCA improved the predictive performance of an ANN model from 0.77 to 0.80 [32].

PCA has been used to convert categorical or nominal variables into numerical variables [24,33,34]. PCA improved the performance of logistic regression in terms of accuracy and sensitivity, based on validation testing [20]. Further, PCA has been used to score social capital, as well as a nominal and ordinal variable [28]. For these reasons, we applied it in developing machine-learning models with high predictive performance. 

#### 2.2.2. Linear Regression

LR models are supervised learning models because they use a linear equation based on specific input values and a machine-learning-based output value [35,36]. LR models are easy to interpret and have low computation costs, although they are generally considered unsuitable for modeling nonlinear data [20] and are prone to bias [28]. Because of its benefits, LR has been consistently applied in developing models to predict waste generation, such as for municipal solid waste [36,37,38,39,40,41]. 

#### 2.2.3. K-Nearest Neighbor

KNN is a supervised learning algorithm that applies distance calculations using training data and a pre-defined k value, and a clustering algorithm to find values nearest to k [42]. KNN has been widely used for regression and classification in various fields due to its simplicity and intuitiveness. Three studies [28,29,37] have applied it to predict municipal solid waste and CDW generation. KNN is considered more suitable for low-dimensional data than data with many input variables [28]. 

#### 2.2.4. Decision Tree

DT algorithms predict the final dependent variable by constructing a complex decision-making process that combines several simple decision-making steps; this non-parametric model is used for both regression analysis and classification [43]. DT learning is a supervised predictive model that connects the observed and target values for each item and approximates a function by applying a series of hierarchical rules [44]. DT algorithms allow easy interpretation, have low computational costs [12], and can be applied to both numerical and categorical data, although their performance is reduced by overfitting. They have been applied to CDW prediction [37,45,46] and municipal solid waste prediction [47,48].

### 2.3. Hyperparameter Tuning

A model’s hyperparameters significantly affect its predictive performance, robustness, and generalizability [49]. We therefore performed HP tuning for the applied algorithms (DT, KNN, and LR) to optimize the models (Table 2). For the KNN model, HP tuning was conducted for the metrics (Euclidean, Manhattan, and Chebyshev), weighting (distance and uniform), and k values (also known as k_neighbors), and the model was optimized at different values depending on the metrics, weights, and k values (Figure 2). For the LR model, we considered ridge, lasso, and elastic net regularization in addition to the original LR, adjusting the regulation strength (alpha value) accordingly (Figure 3). For the DT model, we adjusted the maximum tree depth, minimum number of samples for each leaf, and split criteria to prevent overfitting and optimize the model (Figure 4). 

### 2.4. Model Validation and Evaluation

To verify model performance, we applied leave-one-out cross-validation, a special case of *k*-fold cross-validation; this method can achieve more stable results than *k*-fold cross-validation for small datasets because it uses all samples for testing and training to ensure sufficient sample sizes [50,51,52,53,54,55].

To evaluate DWGR prediction model performance we used the *MAE*, *RMSE*, and *R*^2^:(3)$$MAE=\frac{\sum_{i=1}^{n}\left|y_{i–}-x_{i}\right|}{n}$$
(4)$$RMSE=\sqrt{\sum_{i=1}^{n}\frac{{\left(y_{i–}-x_{i}\right)}^{2}}{n}}$$
(5)$$R^{2}=1-\frac{\sum_{i=1}^{n}{\left(y_{i–}-x_{i}\right)}^{2}}{\sum_{i=1}^{n}{\left(y_{i–}-{\bar{x}}_{i}\right)}^{2}}$$
where $x_{i}$ and $y_{i}$ are, respectively, the observed and predicted quantities of demolition waste generated; ${\bar{x}}_{i}$ is the average of $x_{i}$; and *n* is the number of samples. The performance of the models is considered higher as the *R*^2^ value increases and the *MAE* and *RMSE* values decrease.

## 3. Results

### 3.1. Principal Component Analysis

We selected the thirteen PCs that explained 100% of the variance, with PC1 explaining 33.1% (Figure 5). These were used as new input variables to develop a hybrid machine-learning model for DWGR prediction. The variable coefficients (or eigenvectors) of PCs created by the PCA technique are shown in Table 3.

### 3.2. Input Variable Selection

We developed non-hybrid and hybrid models for DWGR prediction using the seven input variables and 13 PCs (Table 4). Among the non-hybrid models, the model using all seven input variables showed the best predictive performance, highlighting the importance of model testing. To generate the hybrid models, we used all seven input variables and thirteen PCs and selected the input variables that produced the best performance for each hybrid model.

### 3.3. Model Performance

Of the non-hybrid models, the KNN (Chebyshev distance) model performed the worst (*R*^2^ = 0.627, *RMSE* = 194.417, and *MAE* = 133.752), while the DT model performed the best (*R*^2^ = 0.872, *RMSE* = 113.976, and *MAE* = 71.432). The KNN (Euclidean uniform) model achieved intermediate performance (average predicted DWGR = 993.54 kg·m^−2^, *R*^2^ = 0.664, *RMSE* = 184.467, and *MAE* = 127.125) models (Figure 6). The KNN algorithm performed significantly worse than the LR and DT models.

Overall, the hybrid models (*R*^2^: 0.836–0.897) performed significantly better than the non-hybrid models, with the LR and KNN hybrids performing better in terms of *R*^2^, *RMSE*, and *MAE*. The KNN hybrid models performed significantly better than the LR hybrid, while the hybrid PCA−DT model (*R*^2^ = 0.849, *RMSE* = 123.538, *MAE* = 80.472) performed worse than the DT model alone (*R*^2^ = 0.872). The KNN hybrid models achieved the best predictive performance, with Euclidean uniform KNN ranking highest (PCA−KNN (Chebyshev distance): *R*^2^ = 0.872, *RMSE* = 114.005, and *MAE* = 72.255; PCA−KNN (Euclidean uniform): 991.80 kg·m^−2^, *R*^2^ = 0897, *RMSE* = 102.116, and *MAE* = 69.013). Based on these results, we selected the PCA–KNN (Euclidean uniform) model as the optimal model for DWGR prediction.

Compared to previously reported PCA hybrid models [24,31,32], our PCA hybrid models achieved better performance. Previously, the PCA−SVM and PCA−ANN hybrid models performed slightly better (*R*^2^ = 0.78 and *R*^2^ = 0.80, respectively) than SVM and ANN alone (*R*^2^ = 0.75 and *R*^2^ = 0.77, respectively) [32,56]. These findings indicate that the improvement in performance achieved by using PCA varies significantly depending on the algorithm used. Similarly, hybrid models combining CATPCA with SVM, RF, and ANN were tested, with the hybrid models performing significantly better for SVM but worse for RF [24]. 

### 3.4. Optimal Model Performance

Including PCA significantly improved DWGR predictive performance for the KNN algorithm, particularly for the Euclidean uniform method (Figure 7) with the PCA−KNN (Euclidean uniform) achieving more accurate predictions. Although many of the KNN (Euclidean uniform) model predictions fell outside the 20% error margin, most were within these margins. Although the KNN (Euclidean uniform) model’s DWGR predictions differed significantly from the observed values (Figure 8), they appear to provide a good approximation. This is supported by the fact that the KNN (Euclidean uniform) hybrid model has an error rate of 7% compared to the mean of the observed values, while for the non-hybrid model, this was 13%.

### 3.5. Importance of Input Variables in the Optimal Model

We examined the importance of the input variables used in the KNN (Euclidean uniform) and PCA−KNN (Euclidean uniform) models using Pearson’s correlation analysis (Table 5). For the non-hybrid model, the number of floors and the floor area were highly important, in addition to its usage, while location appeared to negatively affect DWGR. For the hybrid model, PC1, number of floors, and floor area were highly correlated with DWGR; PC1 provided a new input variable that significantly affects DWGR predictions. For the hybrid model, PC4 and PC2 negatively affected DWGR. For both the non-hybrid and hybrid models, the number of floors and the floor area were important factors.

## 4. Discussion and Recommendations

This study looked into the development of a hybrid ML model using PCA technology to enhance the performance of the DWGR prediction model for the DT, KNN, and LR algorithms. This study sets itself apart from previous studies that used PCs generated from PCA technology as the only variable input. For instance, previous studies that employed PCA to predict municipal solid and C&D waste generation, such as [3,24,31,32,57], utilized the limited PCs generated through PCA as input variables in the model. Cha et al. [24] developed PCA−SVM and PCA−ANN models using the six PCs generated from the six input variables of the raw data by applying CATPCA and presented the SVM (6PCs) model (*R*^2^ = 0.594) as the best predictive performance model. Lu et al. [3] applied PCA to a multi-linear regression model to predict construction waste generation and developed the PCA–MLR model by converting the five input variables (i.e., population, GDP per capita, total construction output, floor space of newly started buildings, and floor space) into PCA. A study by Lu et al. [3], demonstrated that PCA−MLR (5PCs) with 5 PCs applied revealed better performance than other PCA−MLRs. Noori et al. [31] developed the PCA–SVM model to predict MSW generation and proposed a 6 PCs−SVM model (*R*^2^ = 0.752) as the best prediction model. Noori et al. [32] compared the ANN model and the PCA−ANN model to show an improvement in performance with the PCA−ANN model. Based on that study, the ANN (7 PCs) model (*R*^2^ = 0.80) that applied 7 PCs out of 13 PCs generated the best performance. Shi et al. [57] conducted a study on the factors affecting MSW generation in Africa. This study applied PCA to five input variables (i.e., GDP per capita, geographical location, urbanization rate, legal integrity, and law enforcement time) and showed a high correlation with PC1 in 54 African countries. PC1 had large loads on the three input variables (i.e., GDP per capita, geographical location, and urbanization rate), and these variables were presented as key factors affecting MSW generation in African countries. Sun et al. [56] developed an MSW prediction model by applying PCA to the regression analysis and presented the PCA regression model that applied 3 PCs out of 7 PCs as the best performing model.

As mentioned above, previous studies utilized part or all of the PCs that were generated using PCA technology. On the other hand, our study developed a new hybrid ML model by combining the 13 PCs generated through PCA with the existing set of input variables (i.e., location, usage, structure, roof type, wall type, floor area, and number of floors). Through this method, we found that the KNN algorithm showed better predictive performance results than the LR and DT algorithms when combined with PCA. However, it should be noted that these results reflect the characteristics of the input variables used in this study (i.e., five categorical variables and two numerical variables). Additionally, as shown in Table 4, in order to develop an optimal performance ML prediction model, it is necessary to test the model by structuring various input variable groups using the PCs generated through PCA. In other words, empirical attempts through tests on various input variable sets and application of ML algorithms are vital to developing a hybrid ML model with superior predictive performance by applying PCA.

## 5. Conclusions

In this study, we developed three hybrid PCA−machine-learning models (using the DT, KNN, and LR algorithms) and optimized the hyperparameters to improve DWGR predictive performance. Among the non-hybrid models, DT achieved the best performance and KNN (Chebyshev distance) achieved the worst performance. Including PCA improved the predictive performance of the LR and KNN models, with the KNN (Ensemble uniform) hybrid model achieved the best performance.

It is typically difficult to collect DWGR data in the field and, particularly, sufficient data for machine-learning-model development. Using too few variables, or samples that are too small, limits the development of machine-learning models with high predictive performance. Based on our findings, PCA is effective for developing new input variable sets to supplement low-dimensional input data for DWGR prediction. Therefore, we expect that this method will assist companies and researchers in this field to overcome these obstacles and develop high-performance DWGR prediction models. Follow-up research on machine-learning algorithms, including empirical studies, will further improve the performance of machine-learning models for DWGR prediction.

Here, we include several observable limitations throughout the current study. Firstly, the ML model development process demonstrated in this study is unfavorable from a cost perspective, which warrants the development of a simpler and more cost-effective ML model to secure predictive performance in the future. In addition, the various compositions of demolition waste necessitate a waste management strategy during the development of the DWGR model that considers the different types of demolition waste.

## Figures and Tables

**Figure 1 ijerph-20-03159-f001:**
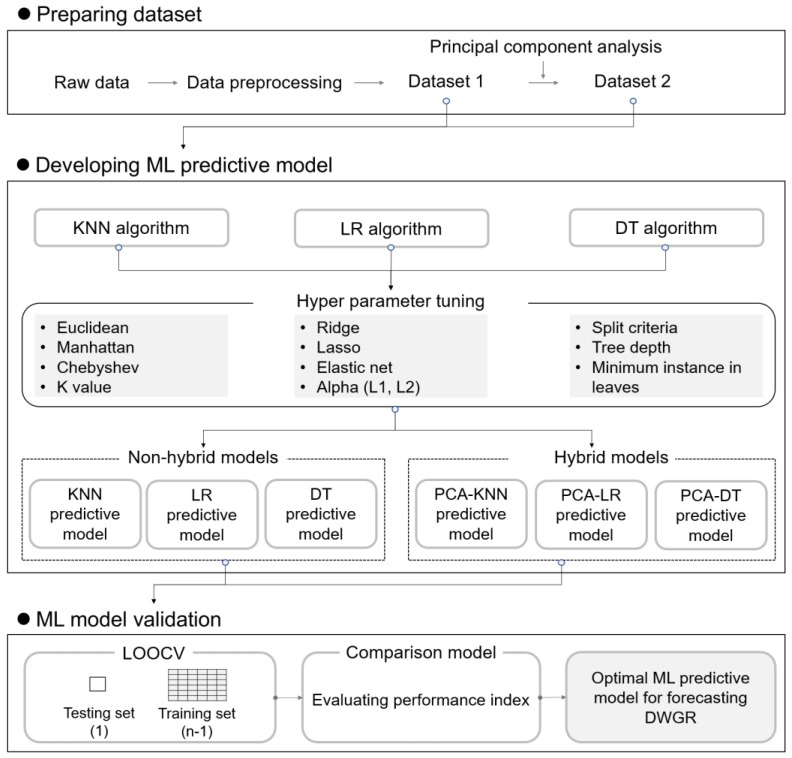
Research workflow used in developing the demolition-waste-generation rate (DWGR) predictive model.

**Figure 2 ijerph-20-03159-f002:**
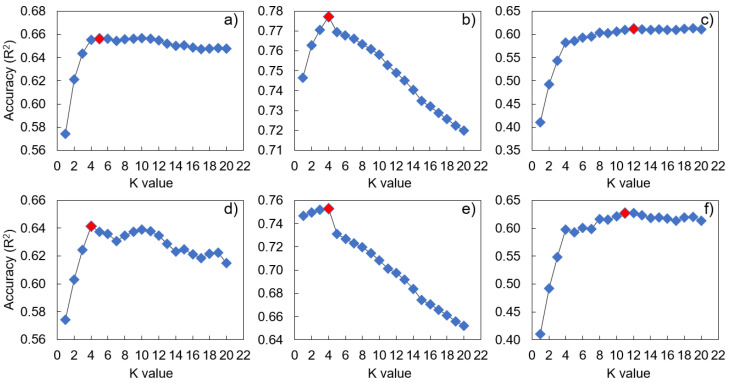
K-nearest neighbor (KNN) model performance according to hyperparameter values. (**a**) Manhattan distance, (**b**) Euclidean distance, (**c**) Chebyshev distance, (**d**) Manhattan uniform, (**e**) Euclidean uniform, and (**f**) Chebyshev uniform. The red diamond represents the hyperparameter value representing the optimal performance.

**Figure 3 ijerph-20-03159-f003:**
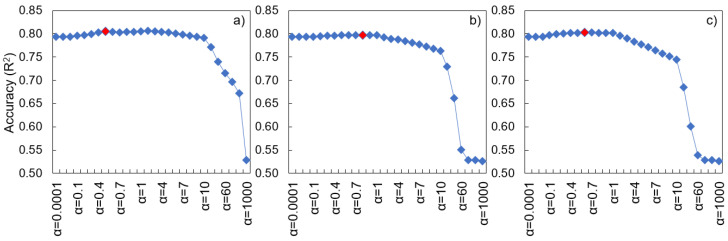
Linear regression model performance according to hyperparameter values. (**a**) Ridge, (**b**) Lasso, and (**c**) Elastic net regularization. The red diamond represents the hyperparameter value representing the optimal performance.

**Figure 4 ijerph-20-03159-f004:**
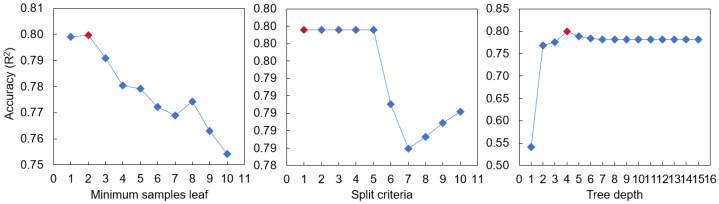
Decision tree model performance according to hyperparameter values (the minimum number of samples for each leaf [minimum samples leaf], split criteria, and maximum tree depth). The red diamond represents the hyperparameter value representing the optimal performance.

**Figure 5 ijerph-20-03159-f005:**
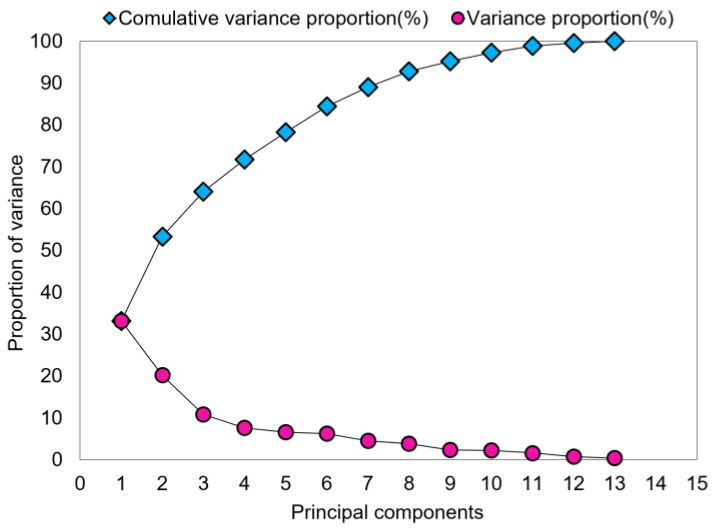
Proportion of variance explained by the number of principal components used.

**Figure 6 ijerph-20-03159-f006:**
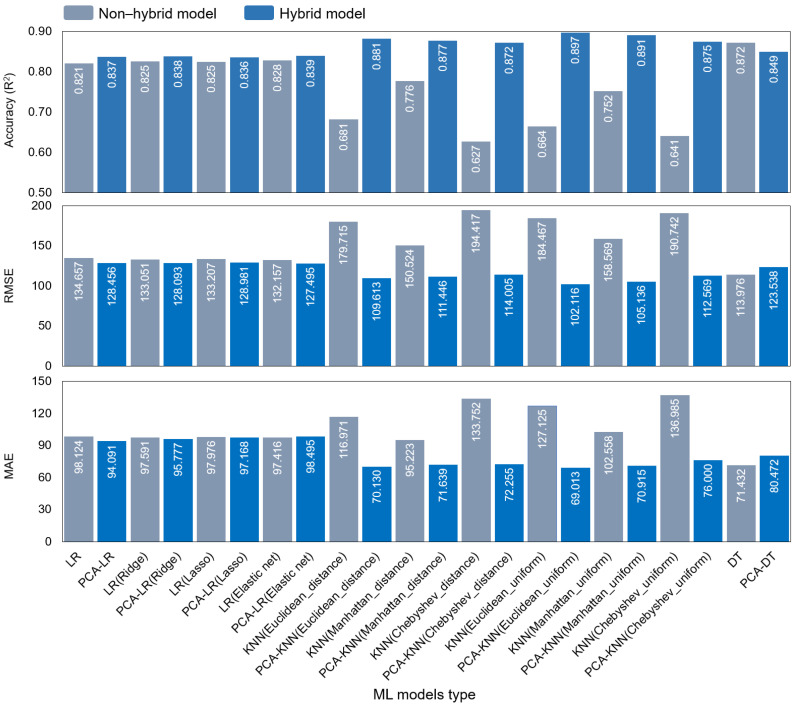
Performance of non-hybrid and hybrid machine-learning models for DWGR prediction.

**Figure 7 ijerph-20-03159-f007:**
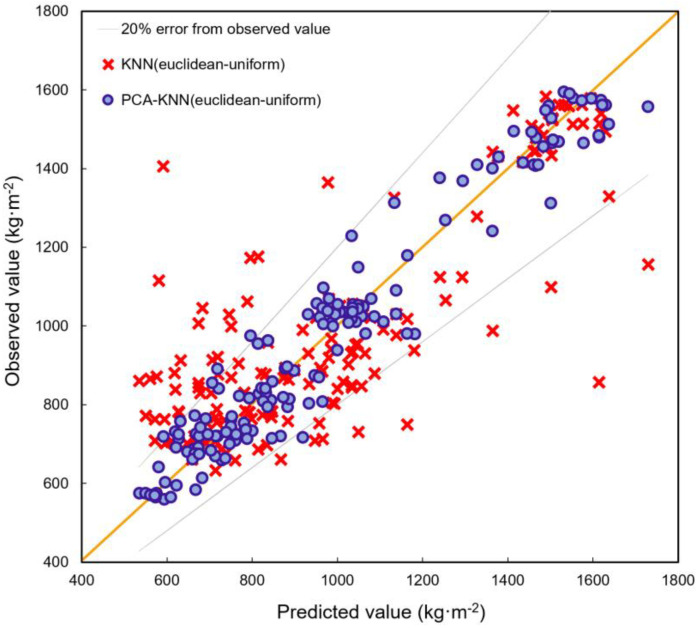
Observed and predicted demolition-waste-generation rate values, comparing the KNN (Euclidean uniform) and PCA−KNN (Euclidean uniform) models. Gray lines: ±20% error on the observed values.

**Figure 8 ijerph-20-03159-f008:**
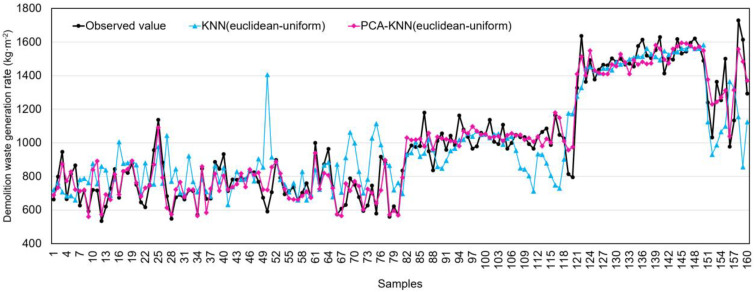
Comparison of prediction results by KNN (Euclidean uniform) and PCA−KNN (Euclidean uniform).

**Table 1 ijerph-20-03159-t001:** Building description and statistical summary of the variables used.

Category		Number of Buildings	Total Demolition Waste Generation (kg)	DWGR (kg·m^−2^)			GFA (m^2^)
				Min	Mean	Max	
Location	Project A	81	6,072,522	534.8	741.6	1137.3	8301.0
	Project B	79	18,194,868	795.2	1238.9	1729.1	13,627.3
Usage	Residential	135	17,875,051	534.8	938.0	1629.4	16,994.4
	Mixed-use (residential & commercial)	25	6,392,340	750.9	1253.0	1729.1	4933.8
Structure	Reinforced Concrete	35	12,573,029	795.2	1430.4	1637.0	8652.1
	Concrete block	81	6,360,974	534.8	854.6	1180.5	7539.8
	Concrete brick	15	3,050,583	717.0	1452.1	1729.1	2500.4
	Wood	29	2,282,804	590.5	755.4	883.0	3236.0
Wall type	Brick	32	4,727,204	590.5	995.9	1729.1	4556.0
	Block	121	18,975,282	534.8	1000.8	1637.0	16,579.0
	Soil	7	564,904	668.0	711.8	759.8	4556.0
Roof type	Slab	37	9,062,146	717.0	1228.3	1729.1	7003.5
	Slab and roofing tile	33	6,939,655	931.0	1213.2	1637.0	5080.8
	Slab & slate	3	963,912	813.5	1310.3	1614.8	719.6
	Slate	13	841,899	534.8	599.7	681.8	1384.9
	Roofing tile	74	6,459,779	580.0	820.8	1576.5	7739.5

**Table 2 ijerph-20-03159-t002:** Hyperparameters considered for developing machine-learning predictive models.

Machine-Learning Algorithms	Hyperparameters				
	Title			Tested Values	Selected
KNN	Euclidean	distance	K neighbors	Range (1, 20)	5
	Manhattan				4
	Chebyshev				12
	Euclidean	uniform			4
	Manhattan				4
	Chebyshev				11
LR	Ridge	alpha		Range (0.0001, 1000)	0.5
	Lasso				0.8
	Elastic net				0.6
DT	Min samples leaf			Range (1, 10)	2
	Split criteria			Range (1, 10)	1
	Max depth			Range (1, 15)	4

KNN: k-nearest neighbor; LR: linear regression; DT: decision-tree; Min samples leaf: the minimum number of samples for each leaf; Max depth: the maximum tree depth.

**Table 3 ijerph-20-03159-t003:** Results of principal component analysis.

Variables	Loading of Variables												
	PC 1	PC 2	PC 3	PC 4	PC 5	PC 6	PC 7	PC 8	PC 9	PC 10	PC 11	PC 12	PC 13
Location_project A	0.29	−0.28	0.00	−0.29	−0.04	0.05	0.02	0.17	0.36	−0.26	0.13	−0.08	0.02
Location_project B	−0.29	0.28	0.00	0.29	0.04	−0.05	−0.02	−0.17	−0.36	0.26	−0.13	0.08	−0.02
Floor area	0.31	−0.10	0.32	0.22	−0.01	−0.11	−0.03	−0.11	−0.22	−0.20	−0.19	−0.45	−0.63
Usage_residential	−0.30	−0.15	0.17	0.01	−0.27	−0.24	0.39	0.21	−0.02	−0.11	−0.11	0.04	0.00
Usage_residential & commercial	0.30	0.15	−0.17	−0.01	0.27	0.24	−0.39	−0.21	0.02	0.11	0.11	−0.04	0.00
Structure_concrete block	−0.24	−0.24	−0.43	−0.02	0.10	0.06	−0.12	0.18	0.11	0.10	−0.22	0.11	−0.43
Structure_concrete brick	0.21	0.27	−0.33	−0.07	−0.13	−0.07	0.06	0.16	−0.51	−0.45	0.05	−0.19	0.29
Structure_reinforced concrete	0.28	−0.19	0.36	0.22	0.06	−0.10	0.03	−0.07	0.12	0.30	−0.27	−0.12	0.49
Structure_wood	−0.14	0.31	0.42	−0.15	−0.10	0.09	0.08	−0.28	0.11	−0.11	0.53	0.14	−0.19
Wall type_block	−0.13	−0.44	−0.01	0.17	0.16	−0.17	−0.15	−0.09	−0.21	−0.10	0.36	0.02	0.09
Wall type_brick	0.17	0.39	−0.16	−0.15	−0.01	−0.01	0.38	−0.17	0.26	0.07	−0.32	−0.02	−0.10
Wall type_soil	−0.07	0.18	0.33	−0.06	−0.32	0.38	−0.44	0.52	−0.08	0.07	−0.15	0.00	0.00
Roof type_roofing tile	−0.27	0.16	0.18	−0.18	0.41	−0.26	−0.24	0.00	0.12	−0.32	−0.24	−0.06	0.09
Roof type_slab	0.29	0.14	−0.13	0.16	−0.17	−0.36	0.03	0.34	0.05	0.40	0.36	0.01	−0.15
Roof type_slab/roofing tile	0.06	−0.33	0.00	−0.41	−0.29	0.29	0.10	−0.35	−0.36	0.19	−0.11	0.07	−0.01
Roof type_slab/slate	0.09	−0.03	0.14	0.15	0.54	0.48	0.48	0.34	−0.18	−0.02	0.09	0.11	−0.05
Roof type_slate	−0.10	−0.02	−0.18	0.61	−0.32	0.37	0.02	−0.18	0.31	−0.31	0.01	−0.07	0.09
Number of floors	0.36	−0.02	0.10	0.15	−0.06	−0.14	−0.10	−0.03	−0.08	−0.27	−0.20	0.82	−0.10

**Table 4 ijerph-20-03159-t004:** Input variables used to develop the non-hybrid and hybrid machine-learning predictive models.

Model		Input Variables
Non-hybrid	LR	location, usage, structure, wall type, roof type, number of floors, floor area
	LR (ridge)	
	LR (lasso)	
	LR (elastic net)	
	KNN (Euclidean distance)	
	KNN (Manhattan distance)	
	KNN (Chebyshev distance)	
	KNN (Euclidean uniform)	
	KNN (Manhattan uniform)	
	KNN (Chebyshev uniform)	
	DT	
Hybrid	LR	PC 1, 2, 3, 4, 6, 8, 9, 10, 11, 13
	LR (ridge)	PC 1, 2, 3, 4, 6, 8, 9, 10, 11, 13
	LR (lasso)	PC 1, 2, 3, 4, 6, 8, 9, 10, 11, 13, location, number of floors
	LR (elastic net)	PC 1, 2, 3, 4, 6, 8, 9, 10, 11, 13, location, number of floors
	KNN (Euclidean distance)	PC 1, 2, location, structure, wall type, floor area
	KNN (Manhattan distance)	PC 1, 2, 4, location, wall type, structure, floor area
	KNN (Chebyshev distance)	PC 1, 2, 4, wall type, structure
	KNN (Euclidean uniform)	PC 1, 2, 4, structure, wall type, floor area, number of floors
	KNN (Manhattan uniform)	PC 1, 2, 4, location, wall type, structure, floor area, number of floors
	KNN (Chebyshev uniform)	PC 1,2,4, location, structure, wall type, floor area
	DT	PC 1, 2, 5, 10, 13

KNN: K-nearest neighbor; LR: linear regression; DT: decision-tree; PC: principal component.

**Table 5 ijerph-20-03159-t005:** Pearson’s correlation analysis of the input variables used to develop the KNN (Euclidean uniform) and PCA−KNN (Euclidean uniform) models.

Model Type	Input Variables	Pearson’s Correlation
KNN (Euclidean uniform)	number of floors	0.782
	floor area	0.747
	usage	0.359
	structure	0.172
	roof type	0.107
	wall type	−0.130
	location	−0.782
PCA−KNN (Euclidean uniform)	PC1	0.783
	number of floors	0.782
	floor area	0.747
	structure	0.172
	PC4	−0.117
	wall type	−0.130
	PC2	−0.377

## Data Availability

All data included in this study are available upon request by contact with the corresponding author.

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
