# Peer review of "Developing a Prediction Model of Demolition-Waste Generation-Rate via Principal Component Analysis"

_ijerph, 2023, doi:10.3390/ijerph20043159_

Round 1

Reviewer 1 Report

Minor comments as below:

1. Introduction section, please highlight the novelty, importance and knowledge gap of this study.

2. Please give GPS information for these two projects(A&B)

3. Conclusion section should be further improved, especially to shorten the length. 

Author Response

Thank you for your review of the completeness of this paper. The revision of this paper reflecting your review is as follows:

Point 1: Introduction section, please highlight the novelty, importance and knowledge gap of this study.

Response 1: Thank you for your hard work in reviewing this thesis. The Introduction has been supplemented by reflecting your opinion. Please refer to lines 90–94.

Point 2: Please give GPS information for these two projects(A&B)

Response 2: Thank you for your review Reflecting your opinion, the related contents have been reinforced in Section 2.1. Please refer to line 105 to 112.

Point 3: Conclusion section should be further improved, especially to shorten the length.

Response 3: Thanks for your comments. The conclusions have been revised and reinforced based on your opinion. Please see the Conclusion section.

Reviewer 2 Report

This paper shows an investigation on the demolition-waste generation-rate predictive modeling via principal component analysis. Overall, this topic is interesting, and this paper can be considered after the following comments being addressed.

(1) There are too many abbreviations in abstract section, which is not friendly for the readers. Part of these abbreviations can be express as full name in abstract section.

(2) Some quantitative data should be added in abstract section.

(3) Most of the references on the waste reutilization are outdated, and more references in recent 3 years should be added in introduction section. The following one reference may be helpful for this work: “Characterization of sustainable mortar containing high-quality recycled manufactured sand crushed from recycled coarse aggregate” (https://doi.org/10.1016/j.cemconcomp.2022.104629)

(4) Some predictive models have been proposed in related literatures. Therefore, the findings in this work and related references should be compared and discussed in the revised manuscript.

(5) Citing references in conclusion section are not suitable, which should be placed in results and discussions. The conclusions should give the findings and results only from this work.  

Author Response

Thank you for your review of the completeness of this paper. The revision of this paper reflecting your review is as follows:

Point 1: There are too many abbreviations in abstract section, which is not friendly for the readers. Part of these abbreviations can be express as full name in abstract section.

Response 1: Thank you for your comments on improving the quality of this manuscript. Reflecting your opinion, the abbreviation was expressed as the full name in the abstract. Please check the revised abstract.

Point 2: Some quantitative data should be added in abstract section.

Response 2: Thank you for your comments on improving the quality of this manuscript. In reflection of your opinion, the contents of quantitative data such as observations and predicted values by models have been supplemented.

Point 3: Most of the references on the waste reutilization are outdated, and more references in recent 3 years should be added in introduction section. The following one reference may be helpful for this work: “Characterization of sustainable mortar containing high-quality recycled manufactured sand crushed from recycled coarse aggregate” (https://doi.org/10.1016/j.cemconcomp.2022.104629)

Response 3: Thank you for your thoughtful comments on this paper. The contents of the introduction have been supplemented by reflecting your opinion. And recent references have been added. Please check lines 49 to 51.

Point 4: Some predictive models have been proposed in related literatures. Therefore, the findings in this work and related references should be compared and discussed in the revised manuscript.

Response 4: I appreciate your opinion and agree. Chapter 4, reflecting your opinion. Discussion and recommendations were added, and in the conclusion part, it was rewritten except for discussion. Please look at Chapters 4 and 5.

Point 5: Citing references in conclusion section are not suitable, which should be placed in results and discussions. The conclusions should give the findings and results only from this work. 

Response 5: As mentioned in the answer to your 4th comment, we newly wrote it except for discussion in the conclusion part. Please refer to the conclusion in Chapter 5 of the paper.

Reviewer 3 Report

The unit for Total in Table 1 should be kg, not kg/m2.

The authors applied PCA with DT, KNN, and linear regression to predict DWGR and improved the prediction accuracy compared to prediction techniques in the previous studies. Utilizing more parameters in a prediction model or combing multiple models should improve prediction accuracy. However, the model complication increases the cost of the prediction analysis. That is why simple prediction models are popular in the field. Moreover, not only DWGR but also the composition of DW is critical to design waste management for C&D. The authors should discuss the pros and cons of the developed prediction technique in terms of data qualities, cost, the future used for demolition waste composition (concrete, wood, metals, etc.)

Author Response

Thank you for your review of the completeness of this paper. The revision of this paper reflecting your review is as follows:

Point 1: The unit for Total in Table 1 should be kg, not kg/m2.

Response 1: thank you for your review The contents of Table 1 have been modified to reflect your opinion. Instead of the existing DWGR, the content was changed to Total demolition waste generation (kg), and the corresponding data values were recalculated. Please refer to Table 1.

Point 2: The authors applied PCA with DT, KNN, and linear regression to predict DWGR and improved the prediction accuracy compared to prediction techniques in the previous studies. Utilizing more parameters in a prediction model or combing multiple models should improve prediction accuracy. However, the model complication increases the cost of the prediction analysis. That is why simple prediction models are popular in the field. Moreover, not only DWGR but also the composition of DW is critical to design waste management for C&D. The authors should discuss the pros and cons of the developed prediction technique in terms of data qualities, cost, the future used for demolition waste composition (concrete, wood, metals, etc.)

Response 2: Thank you for your comments on the completeness of this paper. In response to your comments, we added Chapter 4. Discussion and recommendation, and the contents were modified and reinforced by reflecting your opinions. And the last paragraph of the conclusion was supplemented by reflecting your opinion. Please see Chapter 4. discussion and recommendation and Chapter 5. Conclusions.

Reviewer 4 Report

I have reviewed the paper “Demolition-Waste Generation-Rate Predictive Modelling via Principal Component Analysis”.

This paper offers a hybrid model to predict the demolition waste generation rate by combining principal component analysis with machine learning models. I recommend a few recommendations to the authors: 

1.     I recommend the author revising their title. For example, “Developing a Prediction Model of Demolition-Waste Generation-Rate via Principal Component Analysis”.

2.     The author should mention the originality of this paper. There is no noticeable difference between this research and previous studies.

3.     I am wondering why the author chooses Daegu and Busan for the database. The author may explain the reason.

4.     Table 4 should be put before Fig2.  

5.     The author should better show the principal component load of the principal component analysis.

6.     There are obvious spelling mistakes, such as line 276 and line 289 (number of floors and floor). Authors need to carefully check the manuscript again.

7.     In the conclusions, it is necessary to add the limits of the research. 

Author Response

Thank you for your review of the completeness of this paper. The revision of this paper reflecting your review is as follows:

Point 1:  I recommend the author revising their title. For example, “Developing a Prediction Model of Demolition-Waste Generation-Rate via Principal Component Analysis”.

Response 1: Thanks for your comments. Your suggestion seems more appropriate as a title for this paper. Therefore, the title of the paper has been modified to reflect your opinion.

Point 2: The author should mention the originality of this paper. There is no noticeable difference between this research and previous studies.

Response 1: I appreciate your opinion and agree. Chapter 4, reflecting your opinion. Discussion and recommendations were added, and in the conclusion part, it was rewritten except for discussion. Please look at Chapters 4 and 5.

Point 3: I am wondering why the author chooses Daegu and Busan for the database. The author may explain the reason.

Response 1: Thank you for your review Reflecting your opinion, the related contents have been reinforced in Section 2.1. Please refer to line 105 to 112.

Point 4: Table 4 should be put before Fig2.

Response 1: Thank you for your review. I would be grateful if you could explain the exact reason for this point.

The existing Table 4 is intended to show the difference between the input variables used in the optimal model to which PCA was applied and the input variables used in the model before PCA was applied in this study. As you pointed out, it is not appropriate to enter before Figure 2. It seems not.

Point 5: The author should better show the principal component load of the principal component analysis.

Response 1: Thank you for your review reflecting your opinions, the contents of 3.1 Principal component analysis were reinforced through the main text and Table 3.

Point 6: There are obvious spelling mistakes, such as line 276 and line 289 (number of floors and floor). Authors need to carefully check the manuscript again.

Response 1: Thank you very much for your hard work in carefully reviewing this paper. We sometimes try to reduce these mistakes, but sometimes we are exposed to them. We corrected the part you pointed out, and checked the part for typos and mistakes as a whole.

Point 6: In the conclusions, it is necessary to add the limits of the research.

Response 1: Thank you for your comments on the completeness of the thesis. We reflected your opinion and reinforced the contents by adding the limitations of this study to the conclusion. Please refer to line 373 to 378.

Round 2

Reviewer 4 Report

This is an improved version of the manuscript.

About Point 4:Sorry for the mistake. I had intended to say that Table 2 should be put before Fig2.

Author Response

Point 4: Sorry for the mistake. I had intended to say that Table 2 should be put before Fig2.

Reponse: Thanks again for your review. After reviewing it, it seems more appropriate to reflect your opinion. In terms of the flow of the text, your opinion is correct. Therefore, we rearranged the location of Table 2 to the front of Figure 2. Please refer to Table 2.